# Wafer-scale organic-on-III-V monolithic heterogeneous integration for active-matrix micro-LED displays

Lei Han[1,2], Simon Ogier [3], Jun Li[1,2], Dan Sharkey[3], Xiaokuan Yin[1,2], Andrew Baker[3], Alejandro Carreras[3], Fangyuan Chang[4], Kai Cheng[5] & Xiaojun Guo [1,2] ✉

The organic thin-film transistor is advantageous for monolithic three-dimensional integration attributed to low temperature and facile solution processing. However, the electrical properties of solution deposited organic semiconductor channels are very sensitive to the substrate surface and processing conditions. An organic-last integration technology is developed for wafer-scale heterogeneous integration of a multi-layer organic material stack from solution onto the non-even substrate surface of a III-V micro light emitting diode plane. A via process is proposed to make the via interconnection after fabrication of the organic thin-film transistor. Low-defect uniform organic semiconductor and dielectric layers can then be formed on top to achieve high-quality interfaces. The resulting organic thin-film transistors exhibit superior performance for driving micro light emitting diode displays, in terms of milliampere driving current, and large ON/OFF current ratio approaching $10^{10}$ with excellent uniformity and reliability. Active-matrix micro light emitting diode displays are demonstrated with highest brightness of 150,000 nits and highest resolution of 254 pixels-per-inch.

Recent intensive research and development in micrometer scale III-V semiconductor light emitting diodes (micro-LEDs) shows great promise of creating a technology platform to overcome the limits of existing semiconductor displays for higher brightness, finer resolution, better endurance, more innovative form factors and functions[1–9]. However, a fundamental challenge is efficient and reliable heterogeneous integration of massive arrays of micro-LEDs with transistor pixel circuits for active-matrix (AM) addressing[6,10,11].

Currently, the popularly used architecture, referring to the existing semiconductor display technologies, takes the light-emitting element as the front-plane and the thin-film transistor (TFT) circuit as the back-plane for integration[12]. Due to strict process conditions required for III-V LEDs, direct monolithic heterogeneous integration of III-V LEDs onto the TFT backplanes is not feasible. Therefore, the epitaxially grown LED elements are detached from the sapphire or silicon substrates and then transferred onto the TFT backplanes. Using this type of approach, it is difficult to achieve high resolution arrays with reliable electrical conduction between all the TFTs and LEDs[13–19]. As a result, significant effort on redundancy design and post-process repairing is required[20,21].

The other possible solution is to reverse the structure, but monolithic integration of the TFT array on top of the micro-LED plane also meets a fundamental difficulty. For the industry established inorganic TFTs, a high processing temperature (>350 °C) is needed to form low defect density semiconductor/dielectric layers and interfaces for low leakage current and reliable high current driving[22–24]. Although

[1]School of Electronic Information and Electrical Engineering, Shanghai Jiao Tong University, Shanghai 200240, P. R. China. [2]National Key Laboratory of Science and Technology on Micro/Nano Fabrication, Shanghai Jiao Tong University, Shanghai 200240, P. R. China. [3]SmartKem Ltd., Neville Hamlin Building, Thomas Wright Way, NetPark, Sedgefield TS21 3FG, UK. [4]School of Design, Shanghai Jiao Tong University, Shanghai 200240, P. R. China. [5]Enkris Semiconductor, Inc., Nanopolis Suzhou, 99 Jinji Avenue, Suzhou Industrial Park, Suzhou, Jiangsu province 215124, P. R. China. ✉e-mail: x.guo@sjtu.edu.cn

there were reported low temperature processed metal oxide TFT, the performance and stability were generally degraded[25,26]. The micro-LED pixels require the driving TFT to provide large current in the level from hundreds of microamperes to milliampere, and meanwhile have low enough leakage current to maintain good dark state[1,27]. The high-temperature TFT processes might cause cracks of the multi-layer metal contacts on top of the LED plane, resulting in increase of contact resistance and even failure of electrical connection[28]. The complex pixel circuit composed of at least 2 TFTs and one capacitor (2T-1C) and multi-layer interconnection make the AM TFT array even more difficult to be transferred from the original glass substrate onto the micro-LED plane.

Compared to the inorganic counterparts, the organic TFT (OTFT), using solution processed organic semiconductor (OSC) and dielectric layers, has advantage of forming high-quality channel layer and semiconductor/dielectric interfaces with facile processing at low temperature (<150 °C or even lower)[12,29,30]. Therefore, it could be directly fabricated onto the micro-LED plane without affecting integrity of the underneath layers. However, the large step profiles on the micro-LED plane surface due to presence of thick metal contacts and deep via structures cause difficulties to form thin and uniform OSC and dielectric layers on top by solution processing. Moreover, both the emitted light and the randomly trapped charges at the heterogeneous material interfaces might cause electrical performance fluctuations of the OTFTs.

This work develops an organic-last integration (OLI) technology to effectively address the above challenges of directly making high-performance OTFTs and large-scale circuits directly on top of non-even substrate surface. A thick polymer planarization layer and a metal shielding layer are deposited subsequently. The former is to smoothen the micro-LED plane surface, and the latter is to protect influence of light illumination on the OTFT on top. A "post-OTFT" via process is proposed to make the OTFT-to-micro-LED via interconnection after fabrication of the OTFT, so that the surface is intact before deposition of the OSC layer. Low-defect uniform OSC and dielectric layers can then be formed on top at low temperature (<150 °C) with high-quality metal-semiconductor and semiconductor-dielectric interfaces. The resulting OTFTs exhibit superior performance of milliampere driving current, low leakage current and large ON/OFF current ratio approaching $10^{10}$ with excellent uniformity and reliability. AM micro-LED displays are finally demonstrated, with the highest resolution of 254 PPI and the highest brightness >150k nits, which are the best results so far among all reported TFT-driven AM micro-LED displays[13,14,16–19,31].

## Results

### Organic-last integration structure

Figure 1a illustrates the proposed OLI structure with the OTFT AM array being directly fabricated on top of the micro-LED plane. The micro-LED plane is fabricated on a 4-inch sapphire substrate using the industry standard metal-organic chemical vapor deposition (MOCVD) processes, containing layers of n-doped GaN (N-GaN), multi-quantum-well (MQW), p-doped GaN (P-GaN) and ITO electrode. The OTFT is in a top-gate bottom-contact (TGBC) structure with a metal shielding layer underneath. The detailed process flow is described in Supplementary Fig. 1. To make an AM array, a micrometer-scale thick metal interconnect layer on the micro-LED plane is needed for the power supply ($V_{DD}$) and ground ($V_{SS}$) lines to support large current loading. The large topography of the thick metal layer causes a high risk of electrical shorts at the crossover edges between the power interconnects and the data interconnects on top. To address this issue, a thick polymer planarization layer is deposited onto the micro-LED plane before fabrication of the metal layer for both the data interconnects and the bottom shielding of the OTFTs. The subsequently deposited buffer layer provides a suitable surface energy for uniform crystallization of

the OSC channel and form a high-quality back-channel interface. As shown by the scanning electron micrograph (SEM) in Fig. 1b, the large step (~2.2 μm) is smoothened with this planarization layer for data interconnects on top, which brings excellent electrical insulation properties between the two metal layers at the crossover regions (Supplementary Fig. 2).

To connect the micro-LED contact to the drain of the OTFT, the conventional "pre-OTFT" via process is forming the via interconnection first and the OTFT afterwards (the upper of Fig. 1c). However, due to the thick planarization and buffer layers, formation of deep via structures would interrupt the subsequent crystallization of the OSC layer and continuity of the solution deposited passivation layer on top. Moreover, the plasma-based dry etching for making the via would induce damages to the buffer layer surface and ion charge trapping, which in turn affect the crystallization and charge distribution of the OSC channels, respectively.

A "post-OTFT" via process is developed in this work, where the OTFT-to-micro-LED via is fabricated after fabrication of the OTFT (the bottom of Fig. 1c). Therefore, the buffer layer surface can be maintained to be fully planar and free of ion charge trapping before OSC deposition, and this enables the best OSC crystallization process and uniform electrical characteristics. The passivation layer covers all the device areas after completing the OTFT processes. For the integration, the planarization, buffer, OSC, gate insulator, and passivation layers are all deposited with spin-coating processes. The metal electrodes and interconnects are formed through sputtering and then patterning with photolithography and wet-etching. The via and OSC patterning are completed with dry etching. All the processes can be done with existing III-V micro-LED fabrication facilities. Such low temperature (<150 °C) OLI processes can thus be well compatible with micro-LED processes for direct fabrication of the AM array on top without affecting the micro-LED performance (Supplementary Fig. 3).

### Electrical performance

To drive micro-LED displays, the driving OTFT needs to provide high current in a limited pixel area for both bright luminance and high resolution. For that, in addition to improving the OSC channel mobility, the channel length needs to be minimized for high-density integration and large channel width to length ratio (W/L). Most of previously reported high mobility OSCs used top-contact device structures with long channel lengths (L > 50 μm)[32–37]. Due to the vulnerability of the OSC layers, it is challenging to process fine-resolution metal electrodes on top for shorter channel devices. Moreover, since the channel resistance decreases at shorter channel lengths, the contact resistance is becoming more significant, therefore this is often a bottleneck for further performance improvements in OTFT devices[38–40].

In this work, the TGBC architecture is chosen for its compatibility with fine-resolution metal electrode processes and larger injection area at the contact regions for smaller contact resistance[39,40]. Figure 2a illustrates the structure and the material stack for the fabricated OTFT on top of the micro-LED plane. A blended solution of small molecule OSC and polymer semiconductor binder in tetralin solvent is used to form the channel of high carrier mobility. The small molecule OSCs own merits of highly crystalline structures and purity for high mobility and low trap density, but face challenges of achieving uniform crystallization over large area. Adding polymer semiconductor binder of good miscibility and energy level matching with the small molecule OSC is beneficial for controlling crystallization and obtaining uniform morphology[27,41–43]. As a result, both the device performance and uniformity can be greatly improved compared to that of the devices using the pure small molecule OSC (Supplementary Fig. 4). The self-assembly monolayer (SAM) interfacial layer on the gold source/drain electrodes minimizes the energy barrier for hole injection in ON-state, and suppresses minority charge injection for low leakage current[27,44].

Choosing a proper SAM layer can also help to improve the OSC crystallization on top for high-quality metal-semiconductor interface. Different SAMs are used to modify the contact electrode surface, resulting in various surface morphologies of the OSC layer (Supplementary Fig. 5). The measured $|I_D|$-$V_{GS}$ curves and extracted mobility values of the fabricated devices using different SAMs prove that 3-Fluoro-4-Methoxythiophenol (3-F, 4-OMeBT) is the most suitable SAM for high apparent mobility due to improved crystallization of the OSC film on the contact electrodes (Supplementary Fig. 6). With electrode treatment of the selected SAM, the contact resistance of the devices is significantly reduced (Supplementary Fig. 7).

In the bi-layer structure gate dielectric with overall thickness of around 550 nm, the CYTOP layer forms a non-polar low-trap interface with the OSC layer for high mobility and suppressed charge trapping, and the polymer sputtering resistant layer (SRL) can sustain the gate metal sputtering process and wet etching process on top for low gate dielectric leakage current[45–47]. The measured typical transfer ($|I_D|$-$V_{GS}$) and output ($I_D$-$V_{DS}$) of the OTFT with $L = 3.7\ \mu m$ and $W = 1370\ \mu m$ in a corbino structure are presented in Fig. 2b, showing typical field effect transistor behaviors with negligible hysteresis, large ON-state current of about 1 mA and high ON/OFF ratio near $10^{10}$. The non-ideal output characteristics in the saturation regime is due to relatively small gate dielectric capacitance (about 4.6 nF/cm²), and can be improved by adopting high-$k$ polymer dielectric material for stronger electrostatic

control of the channel from the gate. The negligible contact resistance, as characterized by the transmission line measurement (TLM) method on devices of five channel lengths (Supplementary Fig. 8), enables the device to maintain a high apparent mobility value of 2.6 cm²V⁻¹s⁻¹ at such a short channel for large ON-state current.

The operational stabilities of the OTFTs are vital for AM micro-LED driving. In the pixel circuit, the two type OTFTs are operated in different modes, and thus require different designs. For the pixel driving OTFT, large channel width is needed to provide the required large driving current for micro-LEDs. Therefore, a wide channel ($W = 5.74\ cm$, $L = 6.2\ \mu m$) OTFT is used to characterize its constant current bias stress stability. At a continuous stress current of nearly 1 mA for 4,500 s, the device presents negligible change of the output current (Fig. 2c). A narrower channel OTFT ($W = 200\ \mu m$, $L = 6.2\ \mu m$) is adopted to characterize the operational stabilities for pixel switches. Figure 2d shows that, after continuous dynamic switching for $10^4$ s, the device maintains the high ON/OFF ratio. Negative bias stress (NBS) and positive bias stress (PBS) stability are also characterized, and at both cases the devices show negligible shift after 2 hours continuous bias stress (Supplementary Fig. 9). The excellent operational stabilities of the devices are attributed to the low trap channel interface between the low polar gate insulator (GI) layer and the OSC channel of smooth surface[27]. After being stored in ambient atmosphere with a relative humidity (RH) up to 84% for 55 days, the fabricated OTFT on the micro-

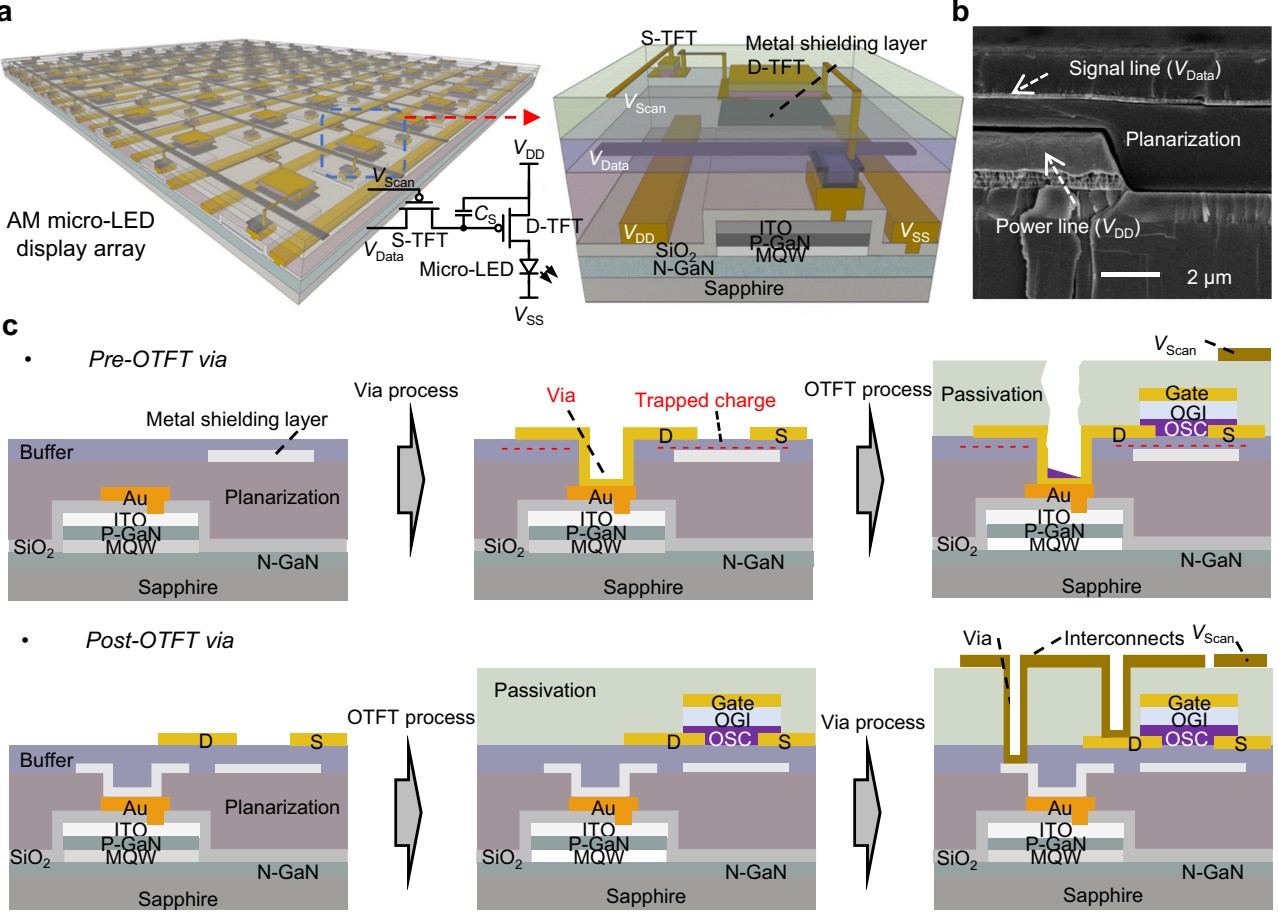

**Fig. 1 | Organic-last integration (OLI) structure. a** Illustration of the proposed OLI structure with the organic thin film transistor (OTFT) active-matrix (AM) array being directly fabricated on top of the micro-LED plane. The pixel driving circuit is based on 2T–1C (T: Transistor C: Capacitor) and contains driver TFT (D-TFT), switch TFT (S-TFT), storage capacitor ($C_S$), micro-LED, signal lines ($V_{Scan}$, $V_{Data}$) and power lines ($V_{DD}$, $V_{SS}$). Specifically, micro-LED is composed of N-doped GaN (N-GaN), multi-quantum-well (MQW), P-doped GaN (P-GaN), indium tin oxide (ITO)

electrode. **b** Scanning electron micrograph (SEM) of the crossover structure between the power line and the data interconnect with a thick polymer planarization layer. **c** Illustration of the advantages with the proposed "post-OTFT" via compared to the conventional "pre-OTFT" via process for electrical connection between the OTFT and the micro-LED, and the OTFT contains source (S)/drain (D) electrodes, organic semiconductor (OSC) layer, organic gate insulator (OGI) layer and gate electrode.

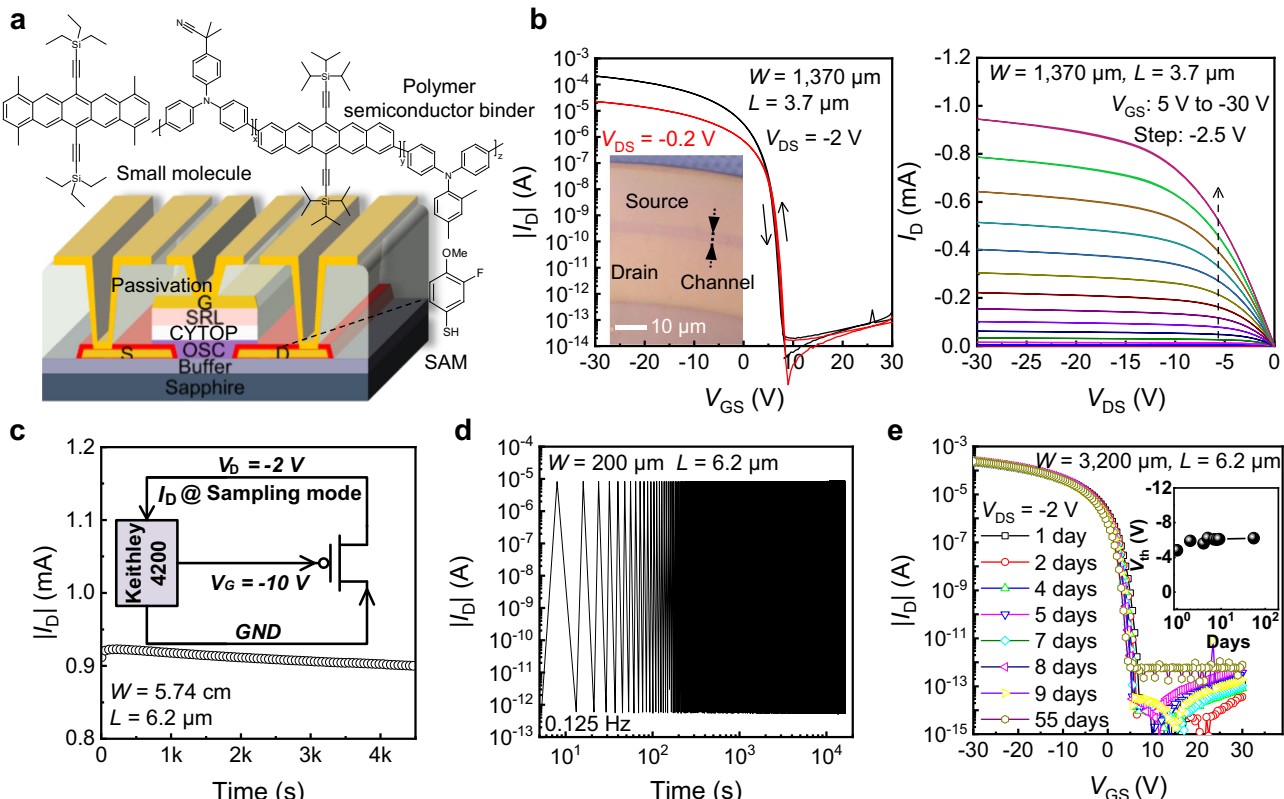

**Fig. 2 | Electrical performance of the fabricated OTFT on top of the micro-LED plane. a** The structure and the material stack for the fabricated OTFT in a top-gate bottom-contact (TGBC) architecture. **b** The measured transfer ($|I_D|$–$V_{GS}$) and output ($I_D$–$V_{DS}$) of the device with channel width ($W$) of 1370 μm and channel length ($L$) of 3.7 μm. **c** The measured evolution of drain current at a continuous stress of nearly 1 mA for 4500 s with a wide channel OTFT ($W$ = 5.74 cm, $L$ = 6.2 μm). **d** The measured ON/OFF current under continuous dynamic switching for $10^4$ s with a narrower channel OTFT ($W$ = 200 μm, $L$ = 6.2 μm). **e** The measured storage stability of device under high relative humidity (RH) atmosphere, indicating the robustness of passivation layer. Inset shows the evolution of threshold voltage ($V_{th}$) during 55 days.

LED plane exhibited negligible change of the measured electrical characteristics (Fig. 2e), indicating the excellent storage stability. After storage for 180 days in the ambient, the slight increased leakage current can recover to the original level with a short-term low-temperature annealing (Supplementary Fig. 10). Such a variation of leakage current doesn't affect the display driving, since it remains in the very small level of $10^{-12}$ A. In practical application, a better encapsulation instead of using a simple thin polymer dielectric layer can be used to protect the intrinsic part from moisture influence.

**Uniformity and scalability**

The measured transfer characteristics ($|I_D|$–$V_{GS}$) for 100 devices on 4-inch size wafer of 3 samples based on the "post-TFT" and "pre-OTFT" via processes are compared in Fig. 3a. The uniformity in terms of the turn-on voltage ($V_{ON}$) and the subthreshold swing ($SS$) are greatly improved with the "post-OTFT" via (Fig. 3b). The large fluctuation of $V_{ON}$ ranging from 6.5 V to 17.5 V with the pre-OTFT via is caused by dry-etching induced random ion charge trapping into the buffer layer. The $SS$ is related to the sub-gap trap states at the channel interface. A large spread of $SS$ values with the "pre-OTFT" via process is ascribed to uncontrollable OSC crystallization caused by the plasma etching process-induced surface roughening of the buffer layer (Supplementary Fig. 11). With the "post-OTFT" process, the buffer layer is intact before fabrication of the OTFT, and thus both charge trapping and surface roughening by the plasma dry etching is suppressed. Specifically, the roughness of buffer layer without RIE process is decreased from 9.70 nm to 0.94 nm, greatly improving uniformity and morphology of OSC layer because the thickness of OSC layer is around

30 nm, proven by the polarized optical micrographs in Supplementary Fig. 12. As a result, the OTFTs can be fabricated onto the micro-LED plane to have reliable electrical connections to the underneath micro-LEDs and achieve excellent uniformity of both $V_{ON}$ and $SS$, which is as well as that with the devices on glass substrate (Supplementary Fig. 13). TCAD simulation also reveals that presence of charge trapping in the buffer layer causes shift of $V_{ON}$, and variation of the sub-gap trap density of states (DOS) causes change of the $SS$ (Fig. 3c). Detailed simulation setting is included in "Methods" section.

With the "post-OTFT" via process, the fabricated OTFTs on the micro-LED plane achieve uniform electrical characteristics, so that the extracted current at the same electrical bias shows excellent linearity with both the channel width ($W$) and the inverse of the channel length ($1/L$) in a wide range (Fig. 3d). Such a scalability brings great convenience for device and circuit designs of different purposes, which require various geometries and dimensions for the current driving and pixel switch OTFTs.

There have been significant efforts on improving the operation frequency of OTFTs by making ultra-scaled channels, reducing the gate to source/drain overlaps and adopting inorganic high-$k$ gate dielectric layers[48–50]. To drive micro-LED displays, achieving large driving current in a limited pixel area is more important. Therefore, the ratio of the apparent mobility to the channel length ($μ/L$) is chosen as the figure-of-merit to evaluate the current driving capability of the device excluding influence the GI capacitance. Moreover, it is preferred to being compatible with existing spin-coating processes for large-area integration on top of the micro-LED plane. Figure 3e compares the $μ/L$ value versus $L$ (representing the capability for high

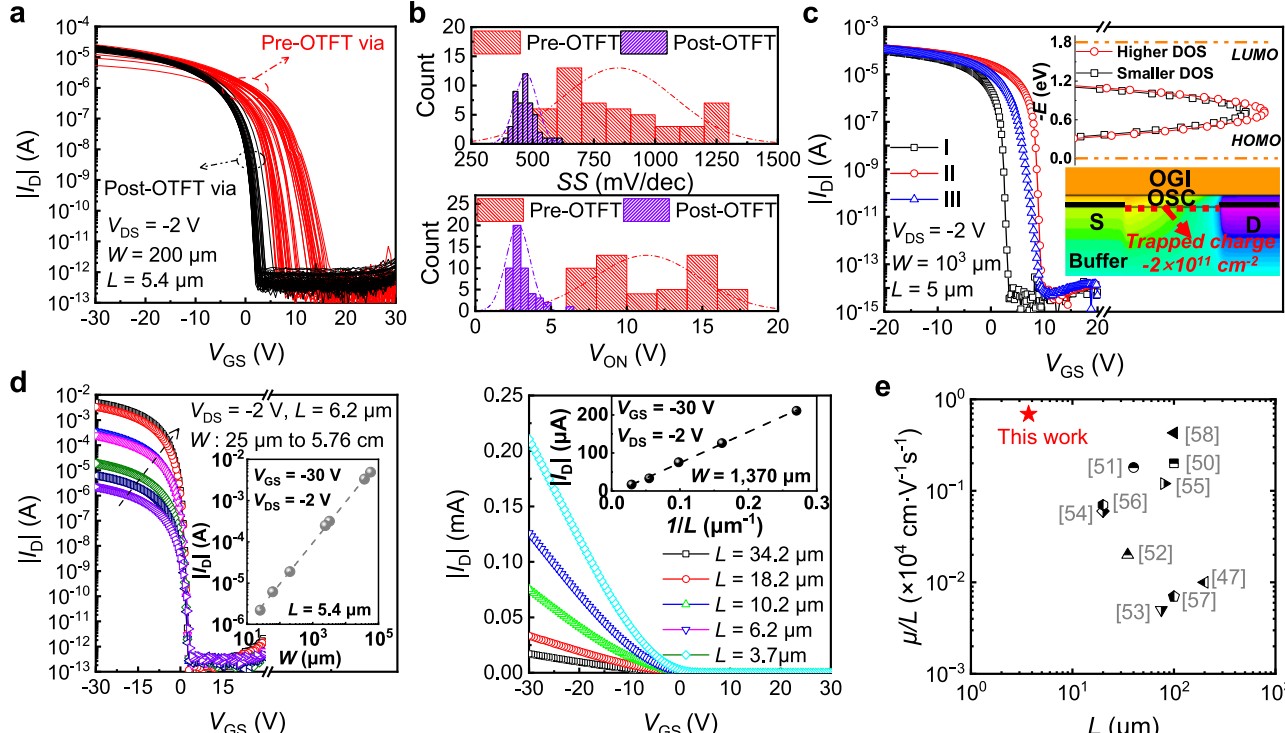

**Fig. 3 | Uniformity and scalability of the fabricated OTFTs based on the post-OTFT via structure. a** The measured transfer characteristics ($|I_D|$ - $V_{GS}$) for 100 devices on 4-inch size wafer of 3 samples based on the "post-TFT" and "pre-OTFT" via processes. **b** The statistical distribution of the extracted turn-on voltage ($V_{ON}$) and subthreshold swing ($SS$) values of the OTFTs. **c** Transfer characteristics ($|I_D|$-$V_{GS}$) obtained using TCAD device simulations on condition of different sub-gap density of state (DOS) profiles and interfacial charge trapping (I: Without interfacial trapped charge and low sub-gap DOS of $10^{15}$ cm$^{-3}$eV$^{-1}$; II: With trapped charge and low sub-gap DOS of $10^{15}$ cm$^{-3}$eV$^{-1}$; III: With trapped charge and high sub-gap DOS of $2 \times 10^{17}$ cm$^{-3}$eV$^{-1}$), which are shown in the inset. **d** Measured transfer characteristics ($|I_D|$-$V_{GS}$) of devices with different channel width ($W$) and channel length ($L$), and linear relationship between the extracted ON-state current (Measured $I_D$ when biased at $V_{GS} = -30$ V and $V_{DS} = -2$ V) and $W/L$. **e** Comparison of $\mu/L$ versus $L$ among reported OTFTs with solution-processed OSC and OGI layers[45, 51–59].

integration density) of the device in this work to previously reported results of OTFTs made of solution-processed OSC and GI layers. The device in this work has the largest $\mu/L$ at the smallest layout footprint for high-resolution integration.

### Display integration

AM micro-LED arrays of different resolution using the 2T-1C pixel design (ranging from 25.4 PPI to 254 PPI) were fabricated on 4-inch sapphire wafer based on the proposed OLI integration approach, as shown in Fig. 4a. The optical micrograph of the fabricated 254 PPI pixel with a pixel size of 100 μm × 100 μm is given in Fig. 4b. The area of the micro-LED is 28 μm × 26 μm, and the driver TFT (D-TFT) with a dimension of 330 μm/2.5 μm occupies an area of 56 μm × 42 μm by using a multi-finger layout design. A maximum driving current close to 0.2 mA in a pixel area of 100 μm × 100 μm, corresponding to a bright luminance near 150k nits (Supplementary Fig. 14), can be reached by modulating the supply voltage ($V_{SS}$) and the data voltage ($V_{Data}$) (Fig. 4c). Figure 4d presents the photo of the 48 × 48 254 PPI AM micro-LED panel being connected to an external driving system through flexible printed circuit (FPC) bonding. Firework and words "O" "L" "I" "L" "E" "D" were dynamically displayed at a frame rate of 60 Hz (Supplementary Movie 1). The snapshots captured from the video are given in Fig. 4e (the bright background is caused by light scattering in the sapphire substrate layer, and can be suppressed by optical structure design). This is the firstly reported high resolution OTFT driven AM micro-LED display in the literature. Certainly, TFTs with a higher mobility like that of metal oxide TFTs or even polycrystalline silicon TFTs in the display industry would be more preferred to driving AM micro-LED displays. However, these inorganic TFTs normally require complex vacuum-based deposition and high-temperature annealing

processes, which would bring significant thermal and mechanical stresses to the underneath layers of micro-LEDs for monolithic integration. Moreover, additional expensive facilities are required to be added to complete the processes. The OTFTs, using low-temperature solution processed OSC and polymer dielectric layers, can have minimal influence to the underneath layers, and only need to adding simple solution coating facilities (e.g. spin-coating). This work shows that, even with an OTFT of mobility less than 3 cm$^2$V$^{-1}$s$^{-1}$, such an OLI approach enables to achieve a display panel of the highest luminance and resolution among the reported TFT-driven AM micro-LED displays (Fig. 4f). By increasing the OSC mobility and gate insulator capacitance, and adopting shorter channel device design, the display performance can be much improved further.

### Discussion

An organic-last-integration (OLI) technology is developed for wafer-scale heterogeneous integration of high-performance OTFTs and high-density active-matrix arrays on non-even substrate surface using solution-based approaches. A "post-OTFT" via process is proposed to form low-defect uniform organic semiconductor and dielectric layers on top to achieve high-quality interfaces. The resulting OTFTs exhibit superior performance for driving micro-LED displays, in terms of milliampere driving current, and large ON/OFF current ratio approaching $10^{10}$ with excellent uniformity and reliability. The first OTFT-driven active-matrix micro-LED displays are demonstrated with highest brightness of 150,000 nits and highest resolution of 254 pixels-per-inch. This work would open a route for realizing active-matrix micro-LED displays through direct fabrication of high-density OTFT arrays onto the micro-LED plane with fully compatible processes. It is expected to apply the similar strategy in this work to higher mobility

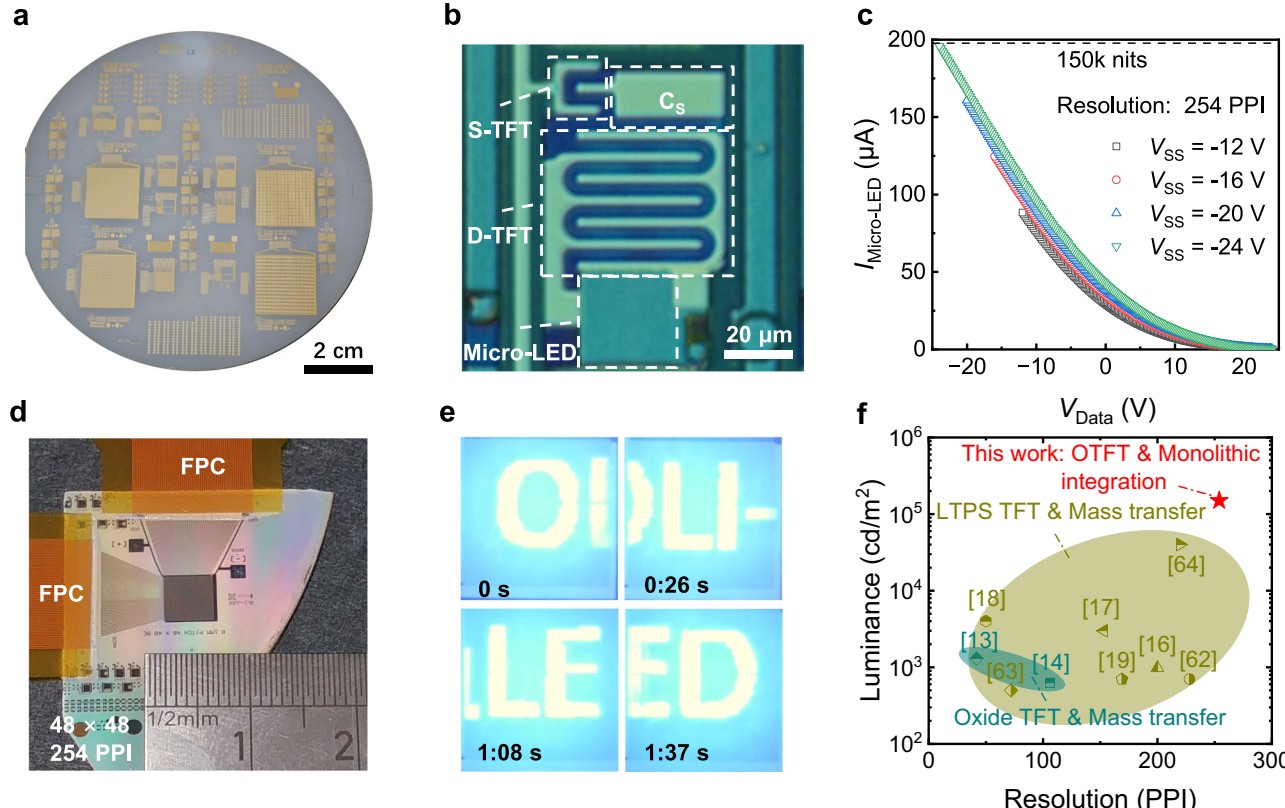

**Fig. 4 | Display integration and demonstration. a** Photo image of the fabricated AM micro-LED arrays on 4-inch sapphire substrate based on the proposed OLI integration approach. **b** Optical micrograph of high-resolution (254 PPI) pixel with multi-finger source/drain design. **c** The measured driving current passing through micro-LED under different voltages for high luminance, with highest luminance of 150,000 nits. **d** Photo image of the 48 × 48 254 PPI AM micro-LED panel being connected to an external driving system through flexible printed circuit (FPC) bonding. **e** The snapshot images captured from the video, which dynamically displayed the words "O" "L" "I" "L" "E" "D" through the 254 PPI AM micro-LED display at a frame rate of 60 Hz. **f** Comparison of the resolution and brightness of the AM micro-LED display using the monolithic OLI approach with previously reported ones based on inorganic TFTs[13, 14, 16–19, 60–62].

small molecule OSCs for continuous improving the device performance, which is worth further investigation. The device performance can also be improved by reducing the channel length and adopting high-*k* dielectric materials to enlarge the gate dielectric capacitance.

## Methods
### Materials for the OTFT
All the layers except for metal layers were based on organic materials by solution process. From bottom to top, acrylate polymer dielectric was used for buffer layer due to its surface energy and reduced roughness. 3-Fluoro-4-Methoxythiophenol (3-F, 4-OMeBT) self-assembled monolayer (SAM) molecule was dissolved in isopropanol to modify the source-drain (S-D) electrodes for smaller injection barrier. OSC ink was composed of small molecule (triethyl(2-{1,4,8,11-tetramethyl−13-[2-(triethylsilyl)ethynyl]pentacen-6-yl}ethynyl)silane) and polymer semiconductor binder dissolved in tetralin at the weight ratio of 0.4%:0.8%:98.8%. Organic gate insulators consist of CYTOP (CTL-809M) and sputter resistant layer (SRL). TRUFLEXTM PV was exploited for passivation layer.

All the electrodes (S-D, top gate, and interconnect) were based on Au except for bottom gate. Mo/Al/Mo was chosen for bottom gate with a larger thickness for better light blocking property.

### Monolithic integration process of OTFT circuit with micro-LED
Following metallization of the thick Au layer, the substrate was re-planarized using a thick layer of TRUFLEX™ planarizer (a UV cross-linkable acrylate polymer) deposited by spin coating and cured under broadband (i,g,h line) UV light in air (4200 mJ/cm²). A light O₂ plasma treatment was conducted to increase surface energy of substrate with high topography for improving wettability of planarization layer onto substrate. Also, a thiol-based adhesion promoter was applied by spin coating in IPA to aid adhesion of the planarizer to the Au areas. Following crosslinking the wafer was baked at 150°C for 60 min. To form vias down to the Au contact pads, a S1813 resist was spin coated on top of the planarizer and baked on a hotplate. It was then exposed using a Heidelberg MLA direct lithography system (405 nm exposure) and developed in 2.38% TMAH developer in water to form the resist mask for the vias. The vias were then formed by dry etching (RIE O₂, Oxford plasma lab 800+) for 7.5 minutes. Following etching the sample was flood exposed to UV and the remaining protective resist removed in TMAH developer.

A Mo/Al/Mo tri-layer was deposited by sputtering for the back-gate layer. It was patterned by photolithography and wet etching (Phosphoric/Acetic/Nitric acid in water) to form the electrodes for the back gate of the transistor. The layer also acts as a light blocking layer to prevent doping of the OSC by light. After this a base-layer dielectric (TRUFLEX™ base-layer) was spin coated and cured in a similar way to the planarizing layer. Following this, the Au (50 nm thick) source-drain (S-D) electrodes were deposited by sputtering and patterned by photolithography and wet etching (KI/I in water). After patterning the source/drain electrode, the sample was exposed to a light plasma treatment to increase the surface energy of the base-layer followed by spin coating of the self-assembled monolayer (SAM) and OSC with baking at 100 °C for 1 minute following the spin coating. The organic gate insulator (OGI) layers were composed of CYTOP and an acrylate polymer-based sputter resistant layer (SRL)[47]. Specifically, CYTOP was

firstly spin coated at the speed of 1,500 rpm for 20 s and baked at 50 °C, 100 °C for 20 s and 60 s, respectively. SRL was spin coated directly onto CYTOP layer at speed of 500 rpm for 10 s and then 1,250 rpm for 180 s and then UV cured using broadband wavelength mercury lamp (4,200 mJ/cm²) under N₂ flow and finally baked at 120 °C for 5 minutes. The measured contact angle of the SRL solution with added surfactant on top of the hydrophobic CYTOP surface is 43°, showing good wetting property[47] (Supplementary Fig. 15). Then the gate metal (GM, 50 nm Au) was deposited by sputtering and patterned by photolithography. Following resist strip (flood-expose-develop) the samples were etched (RIE, O₂/Ar) to remove the areas of OSC/OGI/SRL that were not covered by the gate metal. Etching proceeded into the baselayer but was stopped short of the back-metal layer. Then the passivation layer (TRUFLEX™ PV) was deposited by spin coating and baked at 115 °C for 1 minute. It was crosslinked by UV curing (4,200 mJ/cm²) and then baked for a further 5 minutes at 120 °C. Via holes were patterned in a similar way to the planarization layer, and the etching was continued until the via reached the back-gate layer. In this way the vias could connect back-gate, SD, GM and interconnect metals. Lastly, the interconnect layer (50 nm Au) was deposited by sputtering and patterned using photolithography and wet etching.

## Device characterization

All the electrical test were conducted based on a semi-automatic probe station (Wentworth) connected to a Keithley SCS 4200 parameter analyzer in an air atmosphere.

Constant current bias stability was tested based on the following condition: $V_G$ and $V_D$ were set to be −10 V and −2 V, respectively. Test mode of Keithley SCS 4200 parameter analyzer was changed from sweeping mode to sampling mode with frequency about 1 s and maximum number of 4096 to continuously collect drain current ($I_D$).

Dynamic switching stability was characterized based on sweeping mode and the bias condition was as follows: Gate terminal was set to be 'Voltage list sweep' and the applied voltages were +20 V/-20 V. Drain was biased at −2 V and drain current was collected at each gate voltage.

Ambient stability was tested based on the same device with channel width to length ratio ($W/L$) of 3200 μm/6.2 μm. There was no pre-annealing process before each test and device was stored in dark, atmospheric environment with relative humidity (RH) ranging from 32% to 84% for next test. Test was conducted based on the following conditions: $V_G$ was set to be 'Voltage linear sweep' mode with voltage ranging from +30 V to −30 V and $V_D$ was set to be 'Voltage bias' mode with value of −2 V. Drain current ($I_D$) was measured at each gate voltage ($V_G$) for transfer ($|I_D| \cdot V_{GS}$) characterization.

## Extraction of device parameter

Contact resistance of device is extracted based on transmission line measurement (TLM) method and the mechanism is shown below:

$$R_{Total} = R_C + R_{Channel} = \frac{\partial V_D}{\partial I_D} \tag{1}$$

$$R_C = Intercept(R_{Total}, L) \tag{2}$$

Where $R_{Total}$ represents total resistance in OTFT, equaling to the sum of channel resistance ($R_{Channel}$) and contact resistance ($R_C$), $V_D$ and $I_D$ are the voltage and current at the drain terminal, respectively, $L$ is the channel length. Linear regime was chosen for conducting $R_C$ extraction and $V_D$ was set to be −2 V. Transfer characteristics ($|I_D| \cdot V_{GS}$) of devices with channel length ($L$) ranging from 3.3 μm to 33.8 μm were measured when $V_D$ is set to be −2 V. The extracted total resistance ($R_{Total}$) is normalized by multiplying channel width ($W = 1$ cm). The y-intercept ($L = 0$ μm) of the curves represents the extrapolated normalized contact resistance ($W \cdot R_C$), where channel resistance ($R_{Channel}$) equals to zero.

Linear mobility is extracted based on transconductance ($g_m$) according to the formula below:

$$\mu = \frac{L \cdot g_m}{W C_{diel.} V_D} = \frac{L}{W C_{diel.} V_D} \cdot \frac{\partial I_D}{\partial V_{GS}} \tag{3}$$

Where $L$, $W$, $C_{diel.}$ and $V_D$ represent channel length, channel width, unit area capacitance of dielectric layer and applied drain voltage, respectively.

## AM micro-LED driving

AM micro-LED panels were connected to the display driver system using a FPC interconnect and anisotropic conductive film (ACF), with bonding achieved after aligning and hot-bar bonding (Myachi bonder).

The display driver system was composed of a FPGA control system and 2 × 48-way driver boards to amplify voltages (up to +/24 V). It is controlled by Labview control software. The system can display images or video which are converted to greyscale from.avi files. Frame rate can be controlled up to 120 Hz and individual line times and gate pulse durations can be adjusted to optimize the display.

## Luminance test

2T-1C circuit test structure was designed for luminance test using Spectroscan Spectroradiometer PR670 system. The received light intensity was firstly calculated by multiplying aperture area with the measured luminance and then normalized by real pixel area for extracting the equivalent pixel luminance.

For high luminance test, luminance-current conversion efficiency was firstly measured with excellent linearity (-756 nits/μA) for panel with resolution of 254 PPI, as shown in Supplementary Fig. 14. Then driving current passing through the micro-LED was measured and multiplied by luminance-current conversion efficiency for luminance extrapolation.

## Film characterization

Scanning electron micrograph (SEM) was taken by exploiting ZEISS ULTRA PLASS system with electron energy of 5 kV.

The AFM measurement was done in tapping mode by exploiting a Bruker NanoMan, Dimension V atomic force microscope system.

## Device simulation

Device simulation is based on a commercial numerical device simulator ATLAS vended by SILVACO to verify the increased sub-gap density of state (DOS) and fixed charge carrier on device performance. Principle of device simulation is based on solving the Poisson's equation, the carrier continuity equations, and the drift-diffusion transport equations at each node of user-defined two-dimensional (2D) mesh. The channel length and the channel width are set to be 5 μm and 1000 μm, respectively. Fixed charge with concentration of $−2 \times 10^{11}$ cm⁻² was introduced at buffer/OSC interface by using interface model. The total density of states (DOS) in the bandgap is modeled with two tail bands (donor-like and acceptor-like) using exponential distribution and two deep level bands (acceptor-like and donor-like) using Gaussian distribution. Donor-like deep states with Gaussian distribution is set as variable with different peak concentration (Higher DOS: $C_{peak} = 2 \times 10^{17}$ cm⁻³eV⁻¹ Lower DOS: $C_{peak} = 1 \times 10^{15}$ cm⁻³eV⁻¹).

## Performance comparison

Apparent mobility to channel length ($\mu/L$) is compared among literature-reported OTFTs with both solution-processed OSC and OGI layers. Device materials and key figure-of-merits are also summarized in Supplementary Table 1.

AM micro-LED display driven by TFT technologies is set as standard for comparison in Fig. 4f. The detailed information is included in Supplementary Table 2.

## Data availability

The 1) Electrical performance of the fabricated OTFT on top of micro-LED; 2) Uniformity and scalability of the fabricated OTFTs based on the post-OTFT via structure; 3) Display integration and demonstration data generated in this study are provided in the Supplementary Information/Source Data file. Source data are provided with this paper.

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

## Acknowledgements

We are grateful to Enkris Semiconductor, Inc. for providing epitaxial wafers, Xiangneng Hualei Optoelectronic Corporation for providing LED manufacturing process. We are grateful for the financial support from Shanghai Municipal Commission of Science and Technology Foundation under Grant No. 21511101304 (X.G.) and National Science Fund for Excellent Young Scholars under Grant No. 61922057 (X.G.).

## Author contributions

X.G. conceived of the idea and supervised the work. S.O. and X.G. designed the overall experiments. L.H., D.S., A.B., and A.C. carried out experiments and collected related data. L.H., S.O., and X.G. analyzed all the data and co-wrote the paper. K.C. fabricated the micro-LED wafers. X.Y. and J.L. assisted in checking the pixel design. F.C. designed the structure of OLI integration array. All authors discussed the results and commented on the manuscript.

## Competing interests

The authors declare no competing interests.
