## [Peer Review File · Nature Communications]

Wafer-scale Organic-on-III-V Monolithic Heterogenous Integration for Active-Matrix Micro-LED DisplaysREVIEWER COMMENTS

Reviewer #1 (Remarks to the Author):

In this work, the authors developed a wafer-scale organic-on-III-V monolithic heterogeneous integration for active-matrix micro-LED displays by using 3-D stacking of organic transistors on top of III-V LED. By developing a Post-OTFT via method together with materials selection and surface modifications, organic TFT could maintain high electrical performance after the whole fabrication process. Overall, the work is well, designed, executed and presented.

My only major concern is the benchmark of the device performance, though appears outstanding, might not be comprehensive and accurate. For instance, the works as follows have demonstrated performance of μ/L superior or similar to that of this work, but not listed in Fig. 3e.

(1) Yamamura, Akifumi, et al. "High-Speed Organic Single-Crystal Transistor Responding to Very High Frequency Band." *Advanced Functional Materials* 30.11 (2020): 1909501.

(2) Zschieschang, Ute, et al. "Stencil lithography for organic thin-film transistors with a channel length of 300 nm." *Organic Electronics* 61 (2018): 65-69.

(3) Höppner, Marco, et al. "High-Frequency Operation of Vertical Organic Field-Effect Transistors." *Advanced Science* 9.24 (2022): 2201660.

For the benchmark of luminance vs. resolution in Fig. 4f, this work is listed as (200k cd/m², 254 DPI). However, my understanding from lines 15-24 on p.10 is that the 200k cd/m² and 254 DPI are not achieved in the same device. Rather, 200k cd/m² is only realized in a low-resolution 25.4 DPI display (because a large area is needed for accommodating large OTFTs to provide sufficient driving current), but not in the high-resolution 254 DPI one. Hence the benchmark could be misleading.

Other than the benchmarks, the paper could be strengthened by addressing the comments as follows.

In terms of the semiconductor layer, the author used a blending of small molecule with polymer semiconductor binder. Reviewer is wondering how well these two components can be mixed. Is there any phase separation between them. In addition, compared to using small molecule semiconductor, what are the advantages of adding this binder in terms of the device performance?

In Fig. 2b, the output curve shows unsaturated drain current under high V_{gs}. Reviewer is wondering what reasons lead to this phenomenon.

In Fig. 2e, the author plots several transfer curves using different colors. However, the labeling is missed.

Reviewer is wondering how the fabrication of organic TFT will influence the performance of LED.

From lines 18-39, the authors claim that the main advantage of OTFTs over inorganic TFTs is the low fabrication temperature, which allows monolithic integration of TFT array on top of the LED plane without affecting its performance. However, inorganic TFTs, e.g., metal-oxide, can be fabricated at below 200°C (e.g., Wang, Mengfei, et al. "High performance gigahertz flexible radio frequency transistors with extreme bending conditions." 2019 IEEE International Electron Devices Meeting (IEDM). IEEE, 2019; Yao, Rihui, et al. "Low-temperature fabrication of sputtered high-k HfO₂ gate dielectric for flexible a-IGZO thin film transistors." Applied Physics Letters 112.10 (2018): 103503). Considering metal-oxide TFTs achieve much lower leakage current and higher driving current than OTFTs, what are the advantages to use OTFT for active-matrix displays?

What is the metal shielding layer used for? What roles does it play differently from the buffer layer?

This work achieves excellent contact resistance that allows aggressive scaling down of the OTFTs, which is a big advance in the field. The authors state it is realized by a self-assembly monolayer that minimizes the energy barrier for hole injection. I understand the design rationale is given in the two references (ref. 25, 41); however, they used materials different from this work and did not reach such good performance (i.e., low contact resistance). Could the authors provide (1) details of optimizing the contact; and (2) evidence of lowered-down energy barrier.

It is perhaps not most appropriate to use μ/L as the metric to compare TFT performance (Fig. 3e). Because presumably, while scaling down the TFT length, good mobility could be maintained, by minimizing the contact resistance through making the S/D contact area large. However, this would hurt TFT speed performance due to the large S/D-G overlap capacitance. In this sense, a more appropriate metric to benchmark TFT performance would be unity-current gain frequency, i.e., f_t . It unifies channel length, mobility and overlap length together.

Reviewer #2 (Remarks to the Author):

This manuscript reports the results related to the application of organic semiconductors as micro-LED pixel-driving devices. This work includes interesting results such as forming an organic semiconductor through a low-temperature process on top of an III-V epi-wafer and manufacturing a backplane TFT for the micro-LED operation. The demonstration of the OTFT backplane for micro-LED is certainly noteworthy. However, this work does not offer a compelling demonstration of the unique performance of micro-LED displays and technological breakthroughs. Moreover, organic semiconductors used in this study are well-known materials that have been published in lots of literature. In this work, the authors failed to show fundamental advances in the understanding of organic materials.

It is recommended that the authors submit the manuscript to another journal more directly relevant to the device and fabrication rather than Nature Communications.

Please, improve the quality of the manuscript with the following comments.

1. the geometry of the channel of TFT presented in the paper is presented differently for each figure. The authors should present the data of OTFTs that are used in the backplane.
2. Moreover, the channel width of the TFT used for actual micro-LED operation is very bulky (~mm scale), making it impossible to implement a high-resolution micro-LED display. In fact, the definition of a micro-LED scale is < 50-100 um. The LED with mm scale is called mini-LED. It is a logical contradiction that a transistor of mm size is used as a backplane TFT of a micro-LED. Without the scaling of the OTFT, it is difficult to emphasize the application to a backplane of the micro-LED display.
3. The carrier mobility of OTFTs used in this work has only 2~3 cm²/Vs. It is known that TFTs should possess high mobility over 10 cm²/Vs to operate self-emission devices such as OLED and micro-LED, except for LCD. Pixel switching by simply adjusting the channel size of TFT without improving the characteristics of the semiconductor itself does not mean much to the display industry.

Reviewer #3 (Remarks to the Author):

The paper titled "Wafer-scale Organic-on-III-V Monolithic Heterogeneous Integration for Active-Matrix Micro LED Displays" by Han et al. presents an effective method for integrating OTFT-driven active-matrix micro-LED displays through the "post-OTFT" via structure. The proposed structure improves the process conditions, resulting in reduced surface roughness, charge trap density, and damage during fabrication, thereby enhancing the stability, performance, and integration of OTFTs for controlling micro-LEDs. While the results are impressive and valuable for related research fields, some clarification is necessary before considering publication. My detailed comments are as follows:

1. The experimental results require accompanying statements. For instance, I suggest supplementing the experimental results to demonstrate the higher crystallization of OSC according to the "post-OTFT" via and "pre-OTFT" via process. For the surface roughness of the buffer layer in Fig.S4, I recommend providing a quantitative comparison with specific roughness parameters.
2. In the case of TLM measurement for extracting contact resistance, the measured points of 3 are insufficient. At least 7 points should be required for accurate resistance extraction.
3. Regarding OSC, are there any specific intentions for selecting the composite of small molecule and polymer semiconductor binder as an OTFT channel material? Could the proposed process be compatible with other organic OSC materials?
4. Please supplement the experimental section with the specific material for the sputter-resistant layer. Additionally, denote the dynamic frequency condition in the switching test (Fig.2d).

5. In Fig. 2e, there appears to be a line that looks like dead pixels. What is the origin of this line?

6. It appears that the off-current of the device was dynamically changed and increased in the ambient stability test results (Fig.2e). What is the reason for this change?

7. The authors mention the use of a buffer layer in fabricating the OTFT, but it would be interesting to see the electrical characteristics (mobility, S.S., resistance, hysteresis, etc.) of the OTFT with and without the buffer layer, to understand its impact on the device performance.

8. Due to its hydrophobic nature and low surface energy of Cytop, depositing the SRL on Cytop is considered to be difficult. Could you please provide further clarification on how you were able to deposit the SRL and its specific role in the experiment?

9. There are typos in the title at "Heterogeneous." In Supplementary Fig.S1, the word "OGI" should be changed to "OSC" in process 7 to 8.

10. OTFTs can have variable electrical characteristics depending on vacuum and atmospheric conditions. It would be helpful to know the duration for which the OTFT can operate stably and the results of reliability testing, to gain a better understanding of the limitations of the device.

Reponses to Referees Letter

NCOMMS-23-11029: Wafer-scale Organic-on-III-V Monolithic Heterogeneous Integration for Active-Matrix Micro-LED Displays"

Reviewer#1: In this work, the authors developed a wafer-scale organic-on-III-V monolithic heterogeneous integration for active-matrix micro-LED displays by using 3-D stacking of organic transistors on top of III-V LED. By developing a Post-OTFT via method together with materials selection and surface modifications, organic TFT could maintain high electrical performance after the whole fabrication process. Overall, the work is well, designed, executed and presented.

ANS: Thank you very much for your positive comments of our work.

Question: My only major concern is the benchmark of the device performance, though appears outstanding, might not be comprehensive and accurate. For instance, the works as follows have demonstrated performance of μ/L superior or similar to that of this work, but not listed in Fig. 3e.

(1) Yamamura, Akifumi, et al. "High Speed Organic Single Crystal Transistor Responding to Very High Frequency Band." *Advanced Functional Materials* 30.11 (2020): 1909501.

(2) Zschieschang, Ute, et al. "Stencil lithography for organic thin-film transistors with a channel length of 300 nm." *Organic Electronics* 61 (2018): 65-69.

(3) Hoppner, Marco, et al. "High-Frequency Operation of Vertical Organic Field-Effect Transistors." *Advanced Science* 9.24 (2022): 2201660.

ANS: We are grateful for your suggestions. These references demonstrate great potential of further performance improvement with OTFTs for high frequency operation. Here, we choose reported OTFTs with both solution-processed dielectric and semiconductor layers for benchmarking, considering a preferred advantage of being compatible with existing spin-coating processes for large-scale integration on top of micro-LED. We have added additional description and the references in the revised manuscript (**Page 10: Line 18-21 & 24-25; Reference 59-61**).

Question: For the benchmark of luminance vs. resolution in Fig. 4f, this work is listed as (200k cd/m², 254 DPI). However, my understanding from lines 15-24 on p.10 is that the 200k cd/m² and 254 DPI are not achieved in the same device. Rather, 200k cd/m² is only realized in a low-resolution 25.4 DPI display (because a large area is needed for accommodating large OTFTs to provide sufficient driving current), but not in the high-resolution 254 DPI one. Hence the benchmark could be misleading.

ANS: Thank you for pointing out this. In the manuscript, Fig. 4f shows that the 254 PPI high resolution panel can achieve a brightness about 150k cd/m². We measured the curves of luminance versus current relationship of the micro-LED, and the pixel current versus the bias voltages (V_{SS} and V_{data}) to estimate the achievable brightness. The measurement results for the 254 PPI as below are added in the revised manuscript to show that a brightness of 150k cd/m² was achieved (**Fig. 4d and Supplementary Fig. 15**). Related description is also added in the main text (**Page 1: Line 19; Page**

Luminance test of high resolution (254 PPI) panel: (a) Measured luminance of micro-LED under different driving current. (b) Measured driving current of 2T-1C pixel under different data voltages. (Supplementary Fig. 15).

Question: In terms of the semiconductor layer, the author used a blending of small molecule with polymer semiconductor binder. Reviewer is wondering how well these two components can be mixed. Is there any phase separation between them? In addition, compared to using small molecule semiconductor, what are the advantages of adding this binder in terms of the device performance?

ANS: Thank you for very valuable questions. The small molecule and the polymer semiconductor binder at a weight ratio of 1:2 is well dissolved in a tetralin solvent for mixture. It has been reported that adding polymer binder is beneficial for better controlling the crystallization to achieve uniform morphology, and there is phase separation in the formed solid thin film (*Adv. Funct. Mater.*, 24, 5969-5976 (2014); *Adv. Funct. Mater.*, 26, 1737-1746 (2016); *Adv. Mater.*, 24, 2441-2446 (2012); *Nat. Commun.*, 6, 8598 (2015)). In this work, the small molecule OSC semiconductor is primarily situated at the upper surface for high mobility, while the amorphous polymer being at the bottom to avoid influence of the bottom electrodes on the crystallization process. The measured electrical characteristics of the fabricated OTFTs with and without using the polymer binder are added in the revised manuscript (Supplementary Fig. 4), as shown below. It can be seen that with the polymer binder the devices have higher mobility (μ), steeper subthreshold swing (SS) and lower leakage current with improved uniformity. Related discussions are added in the main text of the revised manuscript (Page 7: Line 3-10; Reference 43-45).

Performance comparison of OTFTs with/without introducing polymer semiconductor binder: (a) Measured transfer ($|I_D|$ - V_{GS}) characteristics of 54 different devices. (b) Statistic distribution of mobility (μ) and subthreshold swing (SS) (**Supplementary Fig. 4**).

Question: In Fig. 2b, the output curve shows unsaturated drain current under high V_{GS} . Reviewer is wondering what reasons lead to this phenomenon.

ANS: Thank you for your valuable suggestion. In this OTFT, the gate insulator layer uses low- k polymer dielectric materials and has overall thickness of 550 nm, resulting in a relatively small gate insulator capacitance of about 4.6 nF/cm². Therefore, for a device with a relatively short channel length of 3.7 μm , the channel current is influenced by the drain field when it is biased in the saturation regime regime. We will work on adopting high- k polymer dielectric material to enlarge the gate insulator capacitance for better saturation characteristics in further work. Related discussions are added in the main text of the revised manuscript (**Page 7: Line 22 & 30-33**).

Question: In Fig. 2e, the author plots several transfer curves using different colors. However, the labeling is missed.

ANS: We are grateful for your helpful advice. The labeling has been added in **Fig. 2e on page 6** in the revised manuscript.

Question: Reviewer is wondering how the fabrication of organic TFT will influence the performance of LED.

ANS: Thank you for your valuable suggestion. After processing the OTFT on top, the LED doesn't show obvious performance variation attributed to the low temperature (< 150 °C) processes. The captured light emission images and the measured current-voltage characteristics of the micro-LED

array sample after thermal aging at 150 °C, 350 °C and 450 °C for 1 hour are given below. Thermal aging at 150 °C doesn't affect the micro-LED performance, and aging at higher temperature does. The micro-LED thermal aging results are added as **Supplementary Fig. 3** in the supporting information, and related discussion is added in the main text (**Page 5: Line 27-28**).

Characterization of the micro-LED performance before and after annealing under different temperature (Supplementary Fig. 3).

Question: From lines 18-39, the authors claim that the main advantage of OTFTs over inorganic TFTs is the low fabrication temperature, which allows monolithic integration of TFT array on top of the LED plane without affecting its performance. However, inorganic TFTs, e.g., metal-oxide, can be fabricated at below 200 °C (e.g., Wang, Mengfei, et al. "High performance gigahertz flexible radio frequency transistors with extreme bending conditions." 2019 IEEE International Electron Devices Meeting (IEDM). IEEE, 2019; Yao, Rihui, et al. "Low-temperature fabrication of sputtered high-k HfO₂ gate dielectric for flexible a-IGZO thin film transistors." Applied Physics Letters 112.10 (2018): 103503). Considering metal-oxide TFTs achieve much lower leakage current and higher driving current than OTFTs, what are the advantages to use OTFT for active-matrix displays?

ANS: We are grateful for your valuable questions. The key advantage of the OTFTs is that facile solution processes can be used to form high quality channel layer and semiconductor/dielectric interfaces at low temperature (< 150 °C or even lower) for high performance and good stability. It brings benefits for direct processing active-matrix arrays on many thermal-sensitive substrates or functional devices using simple processes. For metal oxide TFTs, the standard manufacturing processes in the industry, including sputtering for the channel layer and PECVD for all the dielectric layers, require high temperature (not less than 350 °C) annealing to obtain high quality films and interfaces for high performance and stability. Despite of some excellent work reported on low temperature processed metal oxide TFTs, the device performance and stability are generally degraded and not fully characterized for the display backplane applications. Moreover, for display integration, in addition of the intrinsic device part, multi passivation and interconnect layers are required. Although the performance of OTFTs, especially the mobility, is still much lower than that of the state-of-the-art metal-oxide TFTs, this work is the first report of a complete active-matrix backplane which can be directly fabricated onto the micro-LED. The mobility of the OTFTs can be

further improved by adopting high mobility OSC materials. Related discussions have been added in the revised manuscript (**Page 2: Line 22-24 & 33-35; Page 10: Line 24-25; Reference 25-26**).

Question: What is the metal shielding layer used for? What roles does it play differently from the buffer layer?

ANS: Thanks for the valuable questions. In this work, the metal shielding layer is used for protecting the channel layer from being exposed to light illumination from the micro-LED. The buffer layer is used for providing a suitable surface for uniform crystallization of the OSC channel and form a high quality back-channel interface. Related descriptions are added in the revised manuscript to make it clearer (**Page 3: Line 3-5; Page 5: Line 2-4; Page 13: Line 33-34**).

Question: This work achieves excellent contact resistance that allows aggressive scaling down of the OTFTs, which is a big advance in the field. The authors state it is realized by a self-assembly monolayer that minimizes the energy barrier for hole injection. I understand the design rationale is given in the two references (ref. 25, 41); however, they used materials different from this work and did not reach such good performance (i.e., low contact resistance). Could the authors provide (1) details of optimizing the contact; and (2) evidence of lowered-down energy barrier.

ANS: Thanks for the valuable suggestions. The contact resistance is related to not only the energy level matching, but also the interface properties between the metal contact and the OSC layer. Choosing a proper SAM layer can help to improve the OSC crystallization on top for high-quality metal-semiconductor interface. Different SAMs (4-FBT, PFBT and 3-F, 4-MeOBT) are used to modify the contact electrode surface, resulting in various surface morphologies of the OSC layer on top as characterized by atomic force micrographs (AFM) below. The transfer curves ($|I_D|$ - V_{GS}) for the devices using different SAMs are measured with the apparent mobility values being extracted. The results prove that 3-F, 4-MeOBT is the most suitable SAM (consistent with that in ref. 25) for high apparent mobility due to improved crystallization of the OSC film on the contact electrodes. The contact resistance of the devices with SAM and without SAM is extracted through TLM method for comparison. It clearly shows that with SAM treatment the contact resistance is significantly reduced. The characterization results are added in the revised Supporting Information as **Supplementary Fig. 5-7**. Related discussions are added in the main text (**Page 7: Line 12-21; Supplementary Fig. 5-7**).

Surface roughness of OSC onto electrodes with different SAMs modification: (a-c) Molecule structures of different SAM and corresponding atomic force micrographs (AFM) of OSC onto electrode with different SAM modification. Scale bar: 10 μm . (Supplementary Fig. 5).

Electrical performance of devices with different SAMs modification: (a) Measured transfer characteristics ($|I_D|$ - V_{GS}) and (b) comparison of mobility for devices with different SAM modification. (Supplementary Fig. 6).

Electrical performance of OTFT with/without SAM modification: (a, b) Measured transfer characteristics ($|I_D|$ - V_{GS}) and extracted contact resistance of devices with/without introducing SAM modification. (c) Evolution of contact resistance under different V_{GS} . (d) Comparison of mobility of devices with/without SAM modification. **(Supplementary Fig. 7).**

Question: It is perhaps not most appropriate to use μ/L as the metric to compare TFT performance (Fig. 3e). Because presumably, while scaling down the TFT length, good mobility could be maintained, by minimizing the contact resistance through making the S/D contact area large. However, this would hurt TFT speed performance due to the large S/D-G overlap capacitance. In this sense, a more appropriate metric to benchmark TFT performance would be unity-current gain frequency, i.e., ft. It unifies channel length, mobility and overlap length together.

ANS: We are grateful for your valuable suggestion. Unity-current gain frequency is a crucial figure-of-merit when evaluating TFT operation speed in amplifier circuit design. For display pixel circuits, it is important for the device to achieve large driving current in a limited area. Therefore, the ratio of mobility to channel length (μ/L) and L are selected as the figure-of-merit for performance benchmarking. Related description is strengthened in the revised manuscript to explain it more clearly **(Page 10: Line 18-21).**

Reviewer#2: This manuscript reports the results related to the application of organic semiconductors as micro-LED pixel-driving devices. This work includes interesting results such as forming an organic semiconductor through a low-temperature process on top of an III-V epi-wafer and manufacturing a backplane TFT for the micro-LED operation. The demonstration of the OTFT backplane for micro-LED is certainly noteworthy. However, this work does not offer a compelling demonstration of the unique performance of micro-LED displays and technological breakthroughs. Moreover, organic semiconductors used in this study are well-known materials that have been published in lots of literature. In this work, the authors failed to show fundamental advances in the understanding of organic materials.

It is recommended that the authors submit the manuscript to another journal more directly relevant to the device and fabrication rather than Nature Communications.

ANS: We are grateful for your valuable comments. The importance of this work is to provide a new route for realizing active-matrix micro-LED displays. Currently, various mass transfer techniques are investigated to fabricate the TFT driven active-matrix micro-LED displays. The epitaxially grown LED elements are detached from the sapphire substrates and then transferred onto the TFT backplanes. With this type of methods, it is challenging to achieve high resolution arrays of high yield with reliable electrical conduction between the TFT pixel circuits and LEDs. Based on the merits of OTFTs on low temperature solution processes, this work makes a breakthrough of directly fabricating active-matrix array on top of the micro-LED with processes fully compatible with the LED fabrication. The demonstrated display has the highest resolution and brightness so far among all reported TFT driven active-matrix micro-LED displays.

There has been significant effort on developing high mobility organic semiconductors, which is certainly very important for this field. However, we believe, developing a whole material stack together with the organic semiconductor for high performance OTFT devices and integration is equivalently or even more important. A novel “post-OTFT” via process method is proposed to form uniformly crystallized thin organic semiconductor layer on highly non-smooth substrate surface. The OTFTs fabricated on the micro-LED plane exhibit superior performance of milliamperere driving current, low leakage current and large ON/OFF current ratio with excellent uniformity and reliability. This work well addresses the challenge of making high performance OTFTs and large-scale circuit integration directly on top of non-even substrate surface.

To present the novelty and the importance of the work more clearly, related discussions have been added or emphasized in the revised manuscript (**Page 1: Line 11-12; Page 3: Line 1-2; Page 12: Line 19-26**).

Question: the geometry of the channel of TFT presented in the paper is presented differently for each figure. The authors should present the data of OTFTs that are used in the backplane.

ANS: Thanks for pointing out this. The electrical current of the devices presents excellent linearity versus the channel width (W) and the reciprocal of channel length ($1/L$) attributed to the uniform crystallized channels and negligible contact resistance (Fig. 3d in the manuscript). Therefore, the main conclusions in the manuscript are not affected by the different geometry designs. For different study purposes, we chose different geometry designs. For example, for large current stress test, very wide channel width design is used, and for uniformity test, relatively narrow width design is used. We have added related discussions in the revised manuscript to make it clearly stated (**Page 10:**

Line 16-17).

Question: Moreover, the channel width of the TFT used for actual micro-LED operation is very bulky (~mm scale), making it impossible to implement a high-resolution micro-LED display. In fact, the definition of a micro-LED scale is 50-100 μm . The LED with mm scale is called mini-LED. It is a logical contradiction that a transistor of mm size is used as a backplane TFT of a micro-LED. Without the scaling of the OTFT, it is difficult to emphasize the application to a backplane of the micro-LED display.

ANS: We are grateful for your valuable comments. Although the channel width is large, the layout area of the driving TFT in the pixel is $35\ \mu\text{m}\times 43\ \mu\text{m}$, so a $100\ \mu\text{m}\times 100\ \mu\text{m}$ (254 PPI) can be achieved. Currently, various mass transfer techniques are investigated to fabricate the TFT driven active-matrix micro-LED displays. The epitaxially grown LED elements are detached from the sapphire substrates and then transferred onto the TFT backplanes. With this type of methods, it is challenging to achieve high resolution arrays of high yield with reliable electrical conduction between the TFT pixel circuits and LEDs. This work is to provide a new route for realizing active-matrix micro-LED displays through direct fabrication of the 2T-1C TFT active matrix onto the micro-LED plane. The achieved resolution of 254 PPI with brightness about 150k nits is the best reported result so far for TFT driven active-matrix micro-LED displays. The resolution can be much improved further through not only decreasing minimum feature size, but also increasing the mobility and gate insulator capacitance. Related discussions have been added in the revised manuscript to state the problem and future direction more clearly (**Page 12: Line 4-6 & 14-17 & 26-32**).

Question: The carrier mobility of OTFTs used in this work has only 2~3 cm^2/Vs . It is known that TFTs should possess high mobility over 10 cm^2/Vs to operate self-emission devices such as OLED and micro-LED, except for LCD. Pixel switching by simply adjusting the channel size of TFT without improving the characteristics of the semiconductor itself does not mean much to the display industry.

ANS: Thank you for your valuable comments. We agree that higher mobility is important for active matrix displays to achieve fast refresh rate and high resolution. This work is to provide a new route for realizing active-matrix micro-LED displays through direct fabrication of the 2T-1C TFT active matrix onto the micro-LED plane. The achieved resolution of 254 PPI with brightness of 150k nits is the best reported result so far for TFT driven active-matrix micro-LED displays. A novel “post-OTFT” via process method is proposed to form uniformly crystallized thin organic semiconductor layer on highly non-smooth substrate surface. The OTFTs fabricated on the micro-LED plane exhibit superior performance of milliamperere driving current, low leakage current and large ON/OFF current ratio with excellent uniformity and reliability. This work well addresses the challenge of making high performance OTFTs and large-scale circuit integration directly on top of non-even substrate surface. Related discussions have been added in the revised manuscript (**Page 1: Line 11-12; Page 3: Line 1-2; Page 12: Line 19-26**).

Reviewer#3: The paper titled "Wafer-scale Organic-on-III-V Monolithic Heterogeneous Integration for Active-Matrix Micro LED Displays" by Han et al. presents an effective method for integrating OTFT-driven active-matrix micro-LED displays through the "post-OTFT" via structure. The proposed structure improves the process conditions, resulting in reduced surface roughness, charge trap density, and damage during fabrication, thereby enhancing the stability, performance, and integration of OTFTs for controlling micro-LEDs. While the results are impressive and valuable for related research fields, some clarification is necessary before considering publication.

ANS: Thank you very much for your positive comments of our work.

Question: The experimental results require accompanying statements. For instance, I suggest supplementing the experimental results to demonstrate the higher crystallization of OSC according to the "post-OTFT" via and "pre-OTFT" via process. For the surface roughness of the buffer layer in **Supplementary Fig. 11**, I recommend providing a quantitative comparison with specific roughness parameters.

ANS: We are grateful for your valuable suggestions. The quantitative comparison results of the buffer layer surface roughness are added in **Supplementary Fig. 11** in the revised manuscript, showing that the mean roughness of the buffer layer surface increases from 0.94 nm to 9.70 nm after reactive ion etching. Larger surface roughness affects crystallization quality of the thin OSC layer. The polarized optical micrograph (POM) images of the formed OSC films on rough/smooth buffer layers, proving the improved crystal quality in post-OTFT via structure. We also added the roughness and POM characterization in **Supplementary Fig. 11** and **Supplementary Fig. 12**, respectively (**Page 10: Line 1-5**).

Polarized optical micrograph of OSC onto buffer layer with different roughness. (Supplementary Fig. 12).

Question: In the case of TLM measurement for extracting contact resistance, the measured points of 3 are insufficient. At least 7 points should be required for accurate resistance extraction.

ANS: Thank you for your valuable suggestion. The TLM measurement results based on 5 different channel lengths (L : 3.3 μm , 5.8 μm , 9.8 μm , 17.8 μm and 33.8 μm) are given in the revised manuscript (**Supplementary Fig. 8**). We only put five channel lengths in the design. Since the

results present excellent linearity, we believe the five points measurement can guarantee the extraction accuracy. Related discussions have been added in the revised manuscript (**Page 7: Line 33-34**).

Question: Regarding OSC, are there any specific intentions for selecting the composite of small molecule and polymer semiconductor binder as an OTFT channel material? Could the proposed process be compatible with other organic OSC materials?

ANS: Thanks for the valuable questions. Small molecule organic semiconductor owns advantages for potential of achieving higher mobility, steeper subthreshold swing and lower leakage current attributed to highly crystalline structure and purity. However, it is challenging to form uniform crystallization over large area. Adding the polymer semiconductor binder is helpful to control the crystallization to achieve uniform morphology. Good solubility of the mixture is the basic requirement for selecting the composite. Small molecule organic semiconductors such as TIPS-pentacene, diF-TES ADT and TMTES-Pentacene in this work can all be used. PTAA and other amorphous polymer semiconductors of similar molecule structure and physical-chemical properties have been proved to be suitable choices as the binder. The display performance can be much improved further by adopting high mobility OSC materials. We have added the related discussion in the revised manuscript (**Page 7: Line 4-10; Page 12: 30-32**).

Question: Please supplement the experimental section with the specific material for the sputter-resistant layer. Additionally, denote the dynamic frequency condition in the switching test (Fig. 2d).

ANS: Thank you for your helpful advice. Description of SRL (a type of acrylate polymer) and corresponding patent are added in the experimental section of the revised manuscript and reference section (**Page 14: Line 1; Reference 49**). The dynamic frequency (8 seconds per cycle) used for the switching test was added in **Fig. 2d**.

Question: In Fig. 2e, there appears to be a line that looks like dead pixels. What is the origin of this line?

ANS: Thank you for pointing out this. Two possible reasons might cause this line of dead pixels. Firstly, since all the processes are completed in lab condition, there could be possibility of discontinuity of metal interconnect or via for that line. Secondly, the connection of external driver ICs to the panel is through flexible printed circuit (FPC) bonding, with which there might be some lines are not well connected. By choosing another panel (254 PPI) with better electrical connection, the quality of images and video are much improved in Fig. 4e and **Supplementary Movie 2**. We believe, if using the automatic mass-production facilities in the industry, these problems would be well addressed. Related discussions have been added in the revised manuscript (**Page 12: Line 11-13; Fig. 4d & e; Supplementary Movie 2**).

Question: It appears that the off-current of the device was dynamically changed and increased in the ambient stability test results (Fig.2e). What is the reason for this change?

ANS: We are grateful for your advice. The device used for the measurement uses only simple thin polymer dielectric encapsulation layer. As shown below, after storage for 182 days in the ambient, the leakage current can recover to the original level with a short time low temperature annealing (80 °C for 10 min), proving that the change of the leakage current in the OFF-state comes from moisture. The measurement result is added in the revised manuscript (**Supplementary Fig. 10**). Such a variation of leakage current doesn't affect the display driving, since it remains in the very small level (pA). For practical application, better encapsulation can be used. Related discussions have been added in the revised manuscript (**Page 8: Line 12-17**).

Measured transfer characteristics ($|I_D|$ - V_{GS}) of the same device after 182 days storage by applying a short time annealing. (Supplementary Fig. 10).

Question: The authors mention the use of a buffer layer in fabricating the OTFT, but it would be interesting to see the electrical characteristics (mobility, S.S., resistance, hysteresis, etc.) of the OTFT with and without the buffer layer, to understand its impact on the device performance.

ANS: Thank you for your valuable advice. The buffer layer is used to provide a proper surface property for forming uniformly crystallized OSC channel layer and a high-quality back-channel interface. Discontinuity of OSC film and severe “Coffee ring effect” occur when it is directly deposited onto pure glass or SiO_2 substrate, resulting in the poor yield of device without introducing buffer layer. Therefore, description of the importance of the buffer layer is added (**Page 5: Line 2-4**).

Polarized optical micrograph of OSC onto common substrates without buffer layer: (a) Pure glass substrate. (b) Silicon substrate with 300 nm SiO₂.

Question: Due to its hydrophobic nature and low surface energy of Cytop, depositing the SRL on Cytop is considered to be difficult. Could you please provide further clarification on how you were able to deposit the SRL and its specific role in the experiment?

ANS: Thank you for your valuable question. Surfactant is added in the SRL formulation to aid the wetting on the hydrophobic CYTOP surface. As shown below, the measured contact angle of 43° with the SRL solution on the CYTOP surface shows good wetting properties (that with DI water is 110°). In addition to wetting on CYTOP surface, the SRL helps to form a strong OGI robust to influence of metal sputtering and wet etching processes for making the gate on top. More detailed description of the function of the SRL layer was added in the revised manuscript (**Page 7: Line 25; Page 14: Line 2-4; Reference 49**). The contact angle measurement results were added in the revised manuscript **Supplementary Fig. 16**.

Contact angle of SRL and DI water onto hydrophobic CYTOP layer. (Supplementary Fig. 16).

Question: There are typos in the title at "Heterogeneous." In **Supplementary Fig.1**, the word "OGI" should be changed to "OSC" in process 7 to 8.

ANS: We are grateful for your careful review and have made revisions of all these typos in the revised manuscript (**Page 1: Line 1 & 11; Page 2: Line 11 & 40; Page 1 of S.I.: Line 1;**

Supplementary Fig. 1: Step 7, 8).

Question: OTFTs can have variable electrical characteristics depending on vacuum and atmospheric conditions. It would be helpful to know the duration for which the OTFT can operate stably and the results of reliability testing, to gain a better understanding of the limitations of the device.

ANS: We are grateful for your valuable suggestion. All the electrical characterization were carried out in atmospheric condition, and the device used for the measurement uses only simple thin polymer dielectric encapsulation layer. After storage for 182 days in the ambient, the device maintains the performance with only small change in the OFF-state leakage current. Such a variation of leakage current doesn't affect the display driving, since it remains in the very small level (pA). For practical application, better encapsulation can be used. In addition to excellent operational stability under constant current bias and continuous switching, the devices also present excellent positive bias stress (PBS) and negative bias stress (NBS) stability. The characterization results of storage stability and the PBS/NBS stability are added in the revised manuscript (**Supplementary Fig. 9**). Related discussions are also added in the main text (**Page 8: Line 4-6 & 12-17; Supplementary Fig. 9**).

REVIEWER COMMENTS

Reviewer #1 (Remarks to the Author):

The authors have addressed all my concerns. It is ready for publication.

Reviewer #2 (Remarks to the Author):

The authors have made some efforts to address previous concerns; however, they have not provided satisfactory answers to crucial questions. As a result, this reviewer is unable to grant approval for the publication of this paper.

To effectively drive micro-LED chips smaller than 100 micrometers, the dimensions of the TFTs must be proportionally smaller than the chip size. Unfortunately, the authors have not provided a clear explanation of the TFT dimensions integrated into each LED pixel in the paper. In particular, the TFT width depicted in Figures 2 and 3 exceeds 1300 micrometers (> 1.3 mm scale). This size renders it unsuitable for deployment as a backplane for micro-LED displays.

The configuration shown in Figure 4b lacks persuasiveness, as it involves multiple LEDs connected to a large-area TFT. This arrangement departs from the conventional active matrix structure and raises concerns.

While the paper emphasizes the potential application of backplane TFTs in micro-LED displays, it does not adhere to the established display layout. To operate a micro-LED display efficiently, the dimension and performance of TFT should be comparable to those of IGZO oxide TFTs. It is unreasonable to assume that the utilization of larger TFT sizes to boost output current would suffice for backplane applications.

Reviewer #3 (Remarks to the Author):

The authors have made significant improvements to their work based on the reviewer's comments. However, some issues still need to be addressed before considering publication. The reviewer recommends the following further revisions to the present manuscript:

While the authors have included additional discussions with several references presenting the benefits of blended OSC, there is a lack of clarity in explaining the material's compatibility. The presented

references focus on blended OSC based on diF-TES ADT as a small molecule. The reviewer is concerned about the statement: "The performance can be much improved further by adopting high mobility OSC materials and making shorter channel devices with large gate dielectric capacitance" (Page 12: Lines 30-32). It is suggested to provide more explicit explanations about the compatibility and advantages of the chosen materials in this context.

Several typos have been found in the revised manuscript. These typos, as listed below, should be corrected before publication:

Change "150, 000" to "150,000" (Page 1: Line 19; Page 12: Line 27)

Change "< 150" to "<150" (Page 2: Line 35)

Change "1370" to "1,370" (Page 6: Line 6; Fig. 2a; Page 7: Line 27; Fig. 3d)

Change "3200" to "3,200" (Fig. 2e)

Change "4500" to "4,500" (Page 6: Line 7)

Change "1000" to "1,000" (Page 11: Line 19; Fig. 3c; Page 16: Line 15)

Change "2100" to "2,100" (Page 15: Line 31)

By addressing these points and correcting the mentioned typos, the manuscript will be more suitable for publication.

Point-by-point Response to the Reviewers' Comments
Title: Wafer-scale Organic-on-III-V Monolithic Heterogeneous Integration for Active-Matrix Micro-LED Displays

Reviewer#1: The authors have addressed all my concerns. It is ready for publication.

ANS: Thank you very much for your constructive suggestions and approval of our revision.

Reviewer#2: The authors have made some efforts to address previous concerns; however, they have not provided satisfactory answers to crucial questions. As a result, this reviewer is unable to grant approval for the publication of this paper.

To effectively drive micro-LED chips smaller than 100 micrometers, the dimensions of the TFTs must be proportionally smaller than the chip size. Unfortunately, the authors have not provided a clear explanation of the TFT dimensions integrated into each LED pixel in the paper. In particular, the TFT width depicted in Figures 2 and 3 exceeds 1300 micrometers (> 1.3 mm scale). This size renders it unsuitable for deployment as a backplane for micro-LED displays.

ANS: We are grateful for your valuable comments. For the TFT driven active-matrix micro-LED display pixels, the dimension of the TFT doesn't need to be smaller than that of the micro-LED attributed to high brightness of micro-LEDs. For example, in this work, the area of the micro-LED in the pixel is $28 \mu\text{m} \times 26 \mu\text{m}$. Therefore, for a 254 ppi pixel design, there is much room for placing the TFTs. As shown below, with a multi-finger layout design, the driver TFT (D-TFT) with a dimension of $330 \mu\text{m}/2.5 \mu\text{m}$ occupies an area of $56 \mu\text{m} \times 42 \mu\text{m}$. The wide channel (> 1.3 mm) device in Fig. 2 and 3 is for characterizing the OFF-state leakage current properties and proving that the electrical current of the devices has excellent linearity with the channel width over a wide range. To describe the layout structure more clearly, the pixel layout image as below is added in the revised manuscript (Fig. 4b), and related description has been added in the revised manuscript (Page 11: Line 16 ~ 20).

Layout micrograph of the 254 PPI active-matrix micro-LED pixel (Fig. 4b).

The configuration shown in Figure 4b lacks persuasiveness, as it involves multiple LEDs connected to a large-area TFT. This arrangement departs from the conventional active-matrix structure and

raises concerns.

ANS: Thank you for your constructive comments. The low-resolution pixel design of multiple LEDs is for test purpose. To avoid confusion, we used the layout micrograph of the high-resolution 254 PPI active-matrix micro-LED pixel with one micro-LED in the revised manuscript (Fig. 4b).

While the paper emphasizes the potential application of backplane TFTs in micro-LED displays, it does not adhere to the established display layout. To operate a micro-LED display efficiently, the dimension and performance of TFT should be comparable to those of IGZO oxide TFTs. It is unreasonable to assume that the utilization of larger TFT sizes to boost output current would suffice for backplane applications.

ANS: Thank you very much for the valuable comment. We agree that TFTs with a higher mobility similar to that of IGZO TFTs or even LTPS TFT would be more preferred to driving active-matrix micro-LED displays. However, these inorganic TFTs normally require complex vacuum based deposition and high-temperature annealing processes, which might bring significant thermal and mechanical stresses to the underneath layers of micro-LEDs to achieve such monolithic integration. Moreover, additional expensive facilities are required to be added to complete the processes. The OTFT, using low temperature solution processed organic semiconductor and polymer dielectric layers, can have minimal influence to the underneath layers, and only needs simple solution coating facilities (e.g. spin-coating). This work shows that, even with an OTFT of mobility less than $3 \text{ cm}^2/\text{V}\cdot\text{s}$, such an OLI approach enables to achieve the display panel of the highest luminance ($\sim 150\text{k nits}$) and resolution (254 PPI) among the reported TFT driven active-matrix micro-LED displays. The pixel resolution can be further improved by adopting a higher mobility organic semiconductor, using a finer resolution photolithography process, or enlarging the gate insulator capacitance. Related discussion has been added in manuscript (Page 12: Line 8 ~ 19).

Reviewer#3: The authors have made significant improvements to their work based on the reviewer's comments. However, some issues still need to be addressed before considering publication. The reviewer recommends the following further revisions to the present manuscript:

While the authors have included additional discussions with several references presenting the benefits of blended OSC, there is a lack of clarity in explaining the material's compatibility. The presented references focus on blended OSC based on diF-TES ADT as a small molecule. The reviewer is concerned about the statement: "The performance can be much improved further by adopting high mobility OSC materials and making shorter channel devices with large gate dielectric capacitance" (Page 12: Lines 30-32). It is suggested to provide more explicit explanations about the compatibility and advantages of the chosen materials in this context.

ANS: Thank you very much for your valuable suggestions. Miscibility and energy level matching between the small molecule OSC and the co-polymer binder are the two basic considerations for designing the blended OSC system. In this work, miscibility is improved

by introducing derivative of TMTES-Pentacene into co-polymer molecule structure (Fig 2a). Meanwhile, good energy level matching (HOMO level: -5.19 eV for TMTES-Pentacene and -5.05 eV for co-polymer) is also achieved. As a result, the blended OSC system is helpful to improve crystallization quality and film uniformity, and also achieve high carrier transport mobility. Compared to that of the devices using pure TMTES-pentacene, the mobility of the fabricated devices using the blended OSC system is much increased from $0.28 \pm 0.09 \text{ cm}^2\text{V}^{-1}\text{s}^{-1}$ to $2.55 \pm 0.29 \text{ cm}^2\text{V}^{-1}\text{s}^{-1}$ and the subthreshold swing is decreased from $3.73 \pm 0.49 \text{ V/dec}$ to $0.4 \pm 0.06 \text{ V/dec}$ (Supplementary Fig. 4). There have been several reported small molecule OSCs having high mobility surpassing $10 \text{ cm}^2\text{V}^{-1}\text{s}^{-1}$ (Adv. Funct. Mater., 2022, 32(39): 2202632; Nature, 2011, 475(7356): 364-367; Nat. Commu., 2015, 6(1): 6828). Although the uniformity needs to be significantly improved and the device structures are not feasible for large-scale integration, they show great potential for further performance improvement via tailoring molecule structure. Therefore, it is expected to apply the similar strategy in this work to higher mobility OSCs for improving the device performance, which will be investigated in details in our future work. Related discussions have been added in manuscript (Page 7: Line 7; Page 12: Line 34-38).

Several typos have been found in the revised manuscript. These typos, as listed below, should be corrected before publication:

Change "150, 000" to "150,000" (Page 1: Line 19; Page 12: Line 27)

Change "< 150" to "<150" (Page 2: Line 35)

Change "1370" to "1,370" (Page 6: Line 6; Fig. 2a; Page 7: Line 27; Fig. 3d)

Change "3200" to "3,200" (Fig. 2e)

Change "4500" to "4,500" (Page 6: Line 7)

Change "1000" to "1,000" (Page 11: Line 19; Fig. 3c; Page 16: Line 15)

Change "2100" to "2,100" (Page 15: Line 31)

By addressing these points and correcting the mentioned typos, the manuscript will be more suitable for publication.

ANS: Thank you for your careful review. All these typos have been corrected in the manuscript (Page 1: Line 19; Page 2: Line 27, 35; Page 6: Line 6-7; Page 7: Line 27; Page 11: Line 19; Page 15: Line 31; Page 16: Line 15; Fig. 2a; Fig. 2e; Fig. 3c).